# Nuclear Fusion in Yeast and Plant Reproduction

**DOI:** 10.3390/plants12203608

**Published:** 2023-10-18

**Authors:** Nanami Kobayashi, Shuh-ichi Nishikawa

**Affiliations:** 1Graduate School of Science and Technology, Niigata University, Niigata 950-2181, Japan; snchehon117@gmail.com; 2Faculty of Science, Niigata University, Niigata 950-2181, Japan

**Keywords:** nuclear fusion, female gametogenesis, fertilization, membrane fusion, flowering plants, budding yeast

## Abstract

Nuclear fusion is essential for the sexual reproduction of various organisms, including plants, animals, and fungi. During the life cycle of flowering plants, nuclear fusion occurs three times: once during female gametogenesis and twice during double fertilization, when two sperm cells fertilize the egg and the central cell. Haploid nuclei migrate in an actin filament-dependent manner to become in close contact and, then, two nuclei fuse. The nuclear fusion process in plant reproduction is achieved through sequential nuclear membrane fusion events. Recent molecular genetic analyses using *Arabidopsis thaliana* showed the conservation of nuclear membrane fusion machinery between plants and the budding yeast *Saccharomyces cerevisiae*. These include the heat-shock protein 70 in the endoplasmic reticulum and the conserved nuclear membrane proteins. Analyses of the *A. thaliana* mutants of these components show that the completion of the sperm nuclear fusion at fertilization is essential for proper embryo and endosperm development.

## 1. Introduction

The nucleus contains the genome and functions as the site of essential cellular processes such as DNA replication, transcription, and RNA processing. During the fertilization of various organisms, including animals, plants, and fungi, two nuclei from each of the parents fuse to produce the nucleus of the offspring. This nuclear fusion process, or karyogamy, is essential for reproduction. The nuclear compartment is surrounded by the nuclear envelope, which consists of the inner and outer nuclear membranes. The nuclear membranes play various important roles, including the nucleocytoplasmic transport of macromolecules through the nuclear pore complexes. However, nuclear membranes are barriers to nuclear fusion because they prevent the mixing of materials inside the nuclei.

Nuclear fusion during fertilization is achieved by overcoming the barrier function of the nuclear membranes. In mammalian fertilization, the male and female pronuclei fuse during the first embryonic cell division. The nuclear envelope disassembles by nuclear envelope breakdown (NEBD) when the cells enter the mitotic phase. This allows the mixing of the male and female genomes (Figure 1A) [1]. On the other hand, in many organisms, including plants and fungi, nuclear membranes stay intact during nuclear fusion. In these cases, nuclear fusion is achieved through the fusion of the nuclear membranes of two nuclei (Figure 1B) [2,3].

Recent genetic analyses of *A. thaliana* and live-cell imaging technology have identified the proteins involved in nuclear fusion during plant reproduction. The identified proteins revealed surprising conservations of nuclear fusion mechanisms between yeast and plants. This review aims to provide an overview of the nuclear fusion machinery functioning in plant reproduction and its evolutionary conservations between fungi and land plants.

## 2. Nuclear Fusion in the Sexual Reproduction of Budding Yeast

The mechanism of nuclear fusion during sexual reproduction has been best characterized using budding yeast *Saccharomyces cerevisiae*. Yeast genetics have identified the genes and proteins involved in nuclear fusion. In the sexual phase, the haploid yeast cells of the opposite mating types (*MAT*a and *MAT*α) mate to produce diploid cells (Figure 2A). The haploid cells respond to mating pheromones and exit the mitotic cell cycle [4]. The two cells produce a mating-specific projection and come into contact. Then, the two cells fuse to produce a zygote containing two haploid nuclei. The process of nuclear fusion, called karyogamy, is the final step of yeast mating [2]. Karyogamy consists of two consecutive steps: nuclear congression and nuclear membrane fusion [5]. Yeast mutants defective in karyogamy (Kar^−^) have been isolated and characterized [2]. The zygotes of the mutants defective in the nuclear congression step have the two haploid nuclei positioned far apart in the cells [5]. The mutants defective in the nuclear membrane fusion step produce zygotes in which the two haploid nuclei are juxtaposed but do not fuse [5,6,7]. Genetic and biochemical analyses of karyogamy mutants have identified the components involved in the nuclear membrane fusion process [8,9].

### 2.1. Nuclear Congression

In the nuclear congression process, two haploid nuclei became in close contact in a microtubule-dependent manner [2]. The microtubule from the spindle pole bodies (SPBs), the sole microtubule-organizing center of budding yeast embedded in the nuclear membrane, interconnects the two nuclei, which allows the nuclei to approach each other. Kinesin family motor proteins, Cik1 and Kar3, function during this nuclear movement [10]. Mps3 is a member of the Sad1-UNC-84 homology (SUN) protein family localized in the half-bridge structure of the nuclear membrane adjacent to the core SPB [11,12]. Mps3 interacts with Mps2, an integral membrane protein localized in the outer nuclear membrane, to connect the SPB with the nuclear membrane [13]. Mutants of these proteins are defective in the nuclear congression process. Actin cables also function during the nuclear congression process. Kar9 mediates spindle positioning by connecting microtubules to actin cables [14,15].

### 2.2. Nuclear Membrane Fusion

In the second step, two haploid nuclei fuse to produce a diploid nucleus. In budding yeast, the nuclear membrane stays intact throughout the cell cycle. Nuclear fusion, therefore, must be achieved through the fusion of nuclear membranes, the outer and the inner membranes of the two haploid nuclei. BiP/Kar2, an Hsp70 molecular chaperone in the endoplasmic reticulum (ER) [16], and its partner ER-resident J-domain-containing co-chaperones (J proteins), Jem1/Kar8 [6] and Sec63 [17], were found to facilitate nuclear fusion in karyogamy. Sec63 forms a complex in the ER membrane with Sec71/Kar7 and Sec72 [18]. An electron microscopy examination showed that the Sec63 complex is required for the fusion of the outer nuclear membrane, whereas Jem1/Kar8 functions after the fusion of the outer nuclear membrane [19]. These results suggest that BiP/Kar2 functions in the outer and inner nuclear fusion steps using different J proteins as partners.

Kar5 is a mating-process-specific integral membrane protein that is located adjacent to the SPB [20]. Functional Kar5 orthologs have been identified in zebrafish, the malaria parasite, green algae, and land plants [21,22,23,24]. Kar5 is a member of a conserved protein family functioning in the nuclear membrane fusion process. An electron microscopy examination of the *kar5* mutant zygote suggested that Kar5 is required after outer nuclear membrane fusion and may function in coupling the inner and outer nuclear membranes [25]. Kar5 interacts with the proteins required for the nuclear membrane fusion process. Prm3 is a pheromone-induced peripheral membrane protein on the cytoplasmic face of the outer nuclear membrane, which is required for the outer nuclear membrane fusion step [26]. Kar5 recruits Prm3 adjacent to the SPB. The SPB localization of Kar5 is dependent on Mps3 [25]. Mps3 was identified as a Jem1-interacting protein, and the *mps3* mutant is defective in the nuclear membrane fusion process [12]. These results suggest that Jem1 functions during nuclear membrane fusion by regulating the assembly of Mps3 and, possibly, Kar5 in the SPB as a partner to BiP.

## 3. Nuclear Fusion in Plant Reproduction

During the life cycle of flowering plants, nuclear fusion occurs three times. The first nuclear fusion event occurs during the female gametophyte development [27,28]. Most angiosperm species, including *Arabidopsis thaliana*, have a *Polygonum*-type female gametophyte consisting of one egg cell, two synergid cells, one central cell, and three antipodal cells. A single megaspore produced by meiosis undergoes three rounds of nuclear division cycles, producing an eight-nucleate cell. The subsequent nuclear migration and cellularization result in a seven-celled female gametophyte. The central cell contains two polar nuclei. In *A. thaliana* and other species, the polar nuclei fuse during female gametogenesis to form the secondary nucleus (Figure 2B) [27,28]. The other two nuclear fusion events occur during double fertilization, a process during which two sperm cells released from a pollen tube fertilize two female gametes, an egg, and central cells, producing an embryo and the surrounding endosperm [29,30].

### 3.1. Nuclear Migration

While the nuclear congression step in yeast karyogamy requires microtubules, actin cables play essential roles in the nuclear migration process of a plant reproduction. The disruption of actin filaments in developing female gametophytes influence nuclear migration and alter the final positions of the nuclei [31,32]. The sperm nuclei that have entered the female gametes move toward each female nucleus with the dynamic movement of the actin cables [33]. In in vitro fertilized rice zygotes, the continuous convergence of the actin meshwork toward the egg cell nucleus mediates the migration of the sperm nucleus [34,35]. The disruption of the actin cables in either the egg or the central cell prevents sperm cell nucleus migration. In *A. thaliana*, the involvement of the Rho-GTPase of Plants 8 (ROP8) and class XI myosin in the F-actin dynamics and the sperm nuclear migration in the central cell was reported [36]. A successful sperm nuclear fusion was observed in the female gametophyte of a null allele of *PORCINO* that encodes a subunit of TUBULIN FOLDING COFACTOR C, suggesting that microtubules are dispensable for male gametes’ nuclear migration [33].

### 3.2. Nuclear Membrane Fusion

More than 40 genes were identified through the screening of *A. thaliana* mutants defective in polar nuclei fusion. Since many of the identified genes encode proteins localized in the mitochondria, plastids, plasma membrane, and extracellular region, the nuclear fusion defects in these mutants seemed to be the result of indirect effects [37]. The mutants of transcription factors such as AGL32 and AGL61 showed a polar nuclear fusion defect probably due to defects in their central cell differentiation [38,39]. Recent analyses revealed the conservation of nuclear fusion factors between budding yeast and *A. thaliana*. These include molecular chaperones in the ER and conserved nuclear membrane proteins (Table 1).

#### 3.2.1. Molecular Chaperones in the ER

*A. thaliana* has orthologs of proteins involved in nuclear membrane fusion during yeast karyogamy. The analyses of the mutants of these orthologs showed striking conservations in the mechanisms of nuclear membrane fusion between yeast and plant sexual reproduction. The *A. thaliana* genome encodes three *BIP* genes (*BIP1*, *BIP2,* and *BIP3*). *BIP1* and *BIP2* encode proteins that are 99% identical to each other and are expressed ubiquitously [40]. The female gametophytes harboring the *bip1 bip2* double mutations are defective in polar nuclei fusion [41]. The BiP-deficient female gametophytes contain two unfused polar nuclei in close contact, indicating that the defect is in the membrane fusion step. The third *BIP* gene, *BIP3*, encodes a less well-conserved BiP paralog (80% identical to BiP1 and BiP2) and is expressed only under ER stress conditions in most tissues [40]. BiP3 expression from the *BIP1* promoter fully complemented the polar nuclear fusion defect of the *bip1 bip2* mutant female gametophyte [42,43], indicating that BiP3 has functions that are comparable with those of BiP1 and BiP2.

*A. thaliana* has orthologs of the ER-resident J proteins of yeast [44], and the soluble J proteins ERdj3A, ERdj3B, and P58^IPK^ were shown to function during polar nuclear fusion as partners to BiP [45]. The double mutant female gametophytes lacking P58^IPK^ and ERdj3A or ERdj3B were defective in the polar nuclear fusion. An electron microscopy examination showed that the mutant ovules lacking P58^IPK^ and ERdj3A were defective in the fusion of the outer nuclear membrane. In contrast, in the mutant ovules lacking P58^IPK^ and ERdj3B, the outer nuclear membrane appeared connected via the ER but the inner nuclear membrane remained unfused. These results indicate that P58^IPK^/ERdj3A and P58^IPK^/ERdj3B function at distinct steps of the polar nuclear membrane fusion process as partners to BiP. BiP and ER-resident J proteins likely function during the nuclear fusion process by regulating the conformation or assembly of the proteins required for nuclear membrane fusion. Candidates for such proteins are the nuclear membrane proteins discussed below. BiP and ER-resident J proteins also function during the sperm nuclear fusion at fertilization. The central cell of the *bip1 bip2* or *erdj3a p58^ipk^* double mutant female gametophytes fertilized using wild-type pollen were defective in the sperm nuclear fusion step [46]. By contrast, the central cells of the *erdj3b p58^ipk^* double mutant female gametophytes were not defective in the sperm nuclear fusion step.

#### 3.2.2. Nuclear Membrane Proteins

Recently, two types of nuclear membrane proteins, SUN proteins and GEX1, were shown to function during the nuclear fusion process in *A. thaliana* [24,47]. SUN proteins are integral membrane proteins of the inner nuclear membrane containing a conserved SUN domain. SUN proteins interact with Klarsicht/ANC-1/Syne-1 Homology (KASH) proteins in the outer nuclear membrane through the SUN domain, forming the linkers of the nucleoskeleton and cytoskeleton (LINC) complexes [48]. The *A. thaliana* genome encodes five SUN protein genes. SUN1 and SUN2 are classical SUN proteins with the SUN domain at the C-terminus. SUN3, SUN4, and SUN5 are mid-SUN proteins with internal SUN domains. The expression of a dominant-negative mutant of SUN proteins (SUNDN) in *A. thaliana* pollen was shown to cause the delocalization of the KASH protein, WIP1, from the envelope of the vegetative nucleus, causing defects in the nuclear movement in the pollen tube [49]. The expression of SUNDN in developing female gametophytes resulted in a polar nuclear fusion defect, indicating the roles of SUN proteins in this process [47]. The expression of a SUNDN variant that does not interact with KASH proteins did not cause this polar nuclear fusion defect, suggesting the involvement of a SUN–KASH interaction during polar nuclear fusion. Female gametophytes expressing SUNDN contained unfused polar nuclei in close contact, indicating the observed inhibition at the membrane fusion step. The identification of the KASH proteins functioning in this process awaits further analyses.

The *A. thaliana* gamete-expressed 1 (GEX1) is a functional ortholog of yeast Kar5. The *GEX1* gene was identified by screening the genes expressed in sperm cells [22]. GEX1 is a nuclear membrane protein in the egg and central cell. Time-lapse live-cell imaging using GFP-GEX1 showed that GEX1 expression was detectable first in the central cell, shortly before the polar nuclei were in close contact, and, then, in the egg cell. GEX1-deficient mature female gametophytes were found to contain two unfused polar nuclei in close proximity within the central cell. Electron microscopy showed that the outer membrane of the polar nuclei was connected via the endoplasmic reticulum, whereas the inner membrane remained unfused [24]. The nuclear membrane fusion defect was similar to that observed in the yeast *kar5* mutant zygotes [25]. The sperm nuclear fusion events were defective in the fertilized egg and central cell following the fertilization of the *gex1* female gametophytes with *gex1* pollen. These results indicate that GEX1 is required for all three nuclear fusion events observed in *A. thaliana* reproduction.

The GEX1/Kar5 family proteins have been identified as key factors in nuclear fusion. The *Danio rerio* (zebrafish) Brambleberry (Bmb) is a nuclear membrane protein essential for pronucleus fusion in the zygote and in karyomere fusion during early embryogenesis [21]. The GEX1 orthologs of *Chlamydomonas reinhardtii* and *Plasmodium berghei* are nuclear membrane proteins required for sexual reproduction [23]. Although Kar5, Bmb, and GEX1 differ widely in size and degree of sequence identity, they all contain a well-defined Cys-rich domain (CRD) within their N-terminal region, followed by coiled-coil domains (Figure 3A). While GEX1 contains three transmembrane domains in the C-terminal region, Kar5 and Bmb contain two transmembrane domains. GEX1 orthologs have been identified in various land plants, including eudicots, monocots, basal angiosperms, lycophytes, and bryophytes (Figure 3B) [22,50]. All land plant GEX1 orthologs contain the CRD in their N-terminal region, followed by two or three putative coiled-coil regions and three transmembrane domains [24]. However, sequence identities are not high even between angiosperm GEX1 orthologs; the sequence identity between *Arabidopsis* GEX1 and *Oryza sativa* GEX1 ortholog is lower than 50%. Despite the relatively low sequence identities, the expression of the GEX1 orthologs of *O. sativa,* as well as of *Brassica rapa* from the *Arabidopsis GEX1* promoter, rescued the polar nuclear fusion defect of the *gex1* mutant [51]. In *B. rapa* and rice, fusion starts during female gametogenesis but is not completed before fertilization [52,53]. The completion of polar nuclear fusion in the *gex1* female gametophytes expressing BrGEX1 or OsGEX1 supports that the variation in the times of nuclear fusion completion is not due to differences in the activities of GEX1 orthologs.

## 4. Roles of Nuclear Fusion in Seed Development

The *bip1 bip2* and *erdj3a p58^ipk^* double mutant female gametophytes displayed aberrant endosperm proliferation after fertilization with wild-type pollen [41,45]. Live imaging analyses of endosperm development showed that the aberrant endosperm proliferation was not due to the polar nuclear fusion defect but to the sperm nuclear fusion defect upon fertilization. The fertilized *bip1 bip2* and *erdj3a p58^ipk^* double mutant female gametophytes contained an unfused sperm nucleus in the central cell. Triple nuclear fusion between the unfused sperm nucleus and polar nuclei was achieved during the first endosperm nuclear division. However, the fusion of the sperm nucleus with condensed chromatin resulted in aberrant endosperm nuclear divisions and delayed expression of paternal genes. Aberrant endosperm proliferation was also observed after the fertilization of *gex1* female gametophytes with *gex1* pollen [24].

In contrast, the endosperm proliferated normally after the fertilization of the *erdj3b p58^ipk^* double mutant female gametophytes with wild-type pollen [45]. In the fertilized central cell of the *erdj3b p58^ipk^* double mutant female gametophytes, the sperm nucleus fused with one of the unfused polar nuclei. The unfused polar nuclei fused during the first endosperm nuclear division through nuclear envelope breakdown [46]. Normal seed development was also reported after the fertilization of the *fiona* mutant female gametophytes containing unfused polar nuclei with wild-type pollen [54]. Normal endosperm development was observed in rice, wheat, and maize, producing female gametophytes with unfused polar nuclei [55,56,57]. The fusion of polar nuclei can be omitted in the formation of triploid endosperm.

Sperm nuclear fusion at fertilization seems to be essential for embryo development. The sperm nuclear fusion was defective in the egg cell after the fertilization of the *gex1* female gametophytes with *gex1* pollen [24]. Aberrant embryo development was observed in the resulting *gex1* mutant seeds [24,50]. Analyses of the *gex1* mutant seeds suggested that the first asymmetric cell division occurred after fertilization. However, embryo development was delayed and arrested between the two- and eight-celled embryo stages [24]. This is in good contrast to the fact that embryo development proceeded from the globular to the heart stages after the fertilization of the *erdj3a p58^ipk^* double mutant female gametophytes with wild-type pollen, in which sperm nuclear fusion occurred in the egg cell [45].

## 5. Concluding Remarks

Eukaryotes have evolved unique mechanisms to promote nuclear fusion efficiently during sexual reproduction. The GEX1/Kar5 family proteins are nuclear membrane proteins expressed during the reproduction phase that are conserved between plants, animals, and fungi. The proteins of this family are key factors for nuclear membrane fusion since their deficiency resulted in nuclear fusion defects. The acquisition of this protein family could be one of the critical steps for the establishment of sexual reproduction in eukaryotes. The GEX1/Kar5 family proteins most likely function during the nuclear membrane fusion process alongside other proteins. Analyses using yeast and *A. thaliana* revealed the involvement of ubiquitously expressed proteins such as BiP, ER-resident J proteins, and SUN proteins in nuclear fusion. The involvement of these proteins in the nuclear fusion of other organisms is yet to be analyzed. Other proteins may function during the nuclear membrane fusion step alongside these identified proteins. Future studies will reveal the mechanisms of nuclear membrane fusion at the molecular level.

## Figures and Tables

**Figure 1 plants-12-03608-f001:**
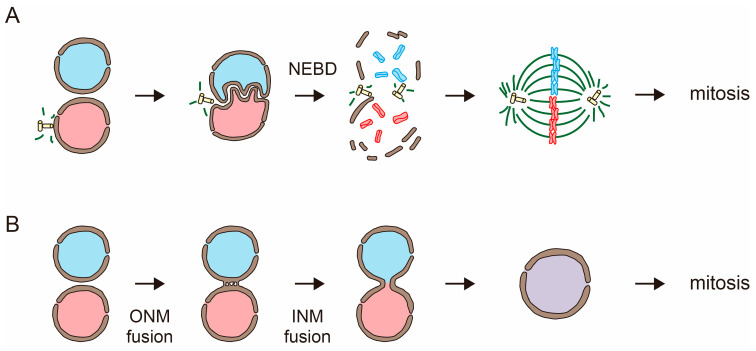
Nuclear fusion in sexual reproduction. (**A**) In mammals, the fusion of pronuclei (red and blue inside) takes place during the first embryonic cell division. Mixing of the male and female genomes is achieved through the nuclear envelope breakdown (NEBD). (**B**) In many organisms, including plants and fungi, the nuclear envelope stays intact during nuclear fusion. The sequential fusions of nuclear membranes, fusion of the outer nuclear membrane (ONM fusion), and fusion of the inner nuclear membrane (INM fusion) result in the production of a diploid nucleus.

**Figure 2 plants-12-03608-f002:**
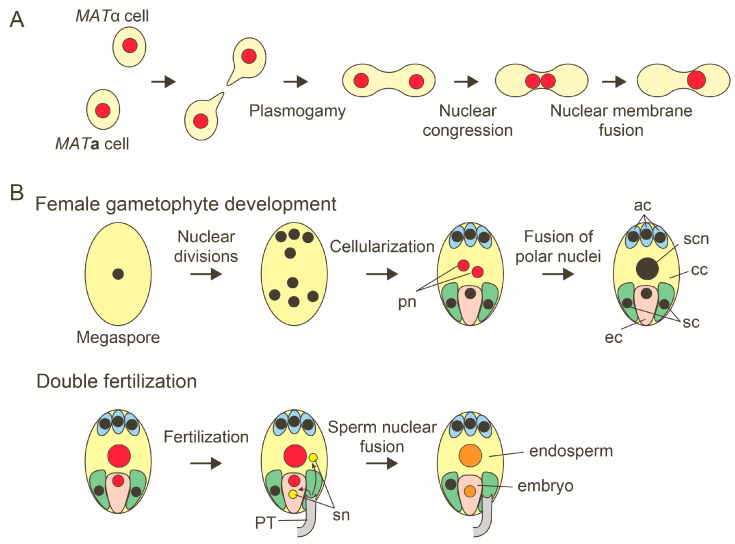
Nuclear fusion in yeast mating and *A. thaliana* lifecycle. (**A**) In the sexual phase, two haploid cells of the opposite mating types produce a mating projection. A zygote is produced by the fusion of two haploid cells (Plasmogamy). In the zygote, two haploid nuclei move to close contact (Nuclear congression) and fuse (Nuclear membrane fusion). (**B**) Three nuclear fusion events in the reproduction of flowering plants. An eight-nucleated and seven-celled female gametophyte is produced from a haploid megagametophyte. The central cell contains polar nuclei that fuse to form a diploid secondary nucleus in *A. thaliana* and other species. During double fertilization, two sperm cells released from a pollen tube fertilize the egg and central cells, producing the embryo and endosperm, respectively. Sperm nuclear fusions take place in the fertilized egg and central cells. pn, polar nuclei; ec, egg cell; sc, synergid cell; cc, central cell; ac, antipodal cell; scn, secondary nucleus; PT, pollen tube; and sn, sperm nuclei.

**Figure 3 plants-12-03608-f003:**
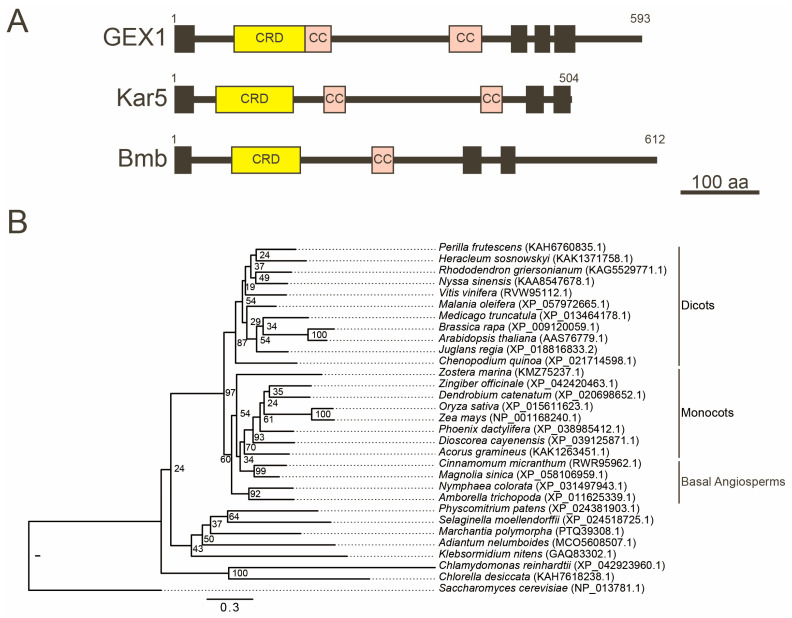
(**A**) Schematic representation of GEX1, Kar5, and Brambleberry (Bmb). GEX1 has one predicted N-terminal signal sequence (black box), one Cys-rich domain (CRD), two coiled-coil domains (CC), and three transmembrane domains (black boxes). Kar5 and Bmb have two transmembrane domains. (**B**) Phylogenetic analysis of GEX1 in Viridiplantae. The maximum-likelihood phylogenetic analysis was performed using the amino acid sequences of GEX1 in Viridiplantae. The species and accession numbers of the GEX1 sequences are shown. *Saccharomyces cerevisiae* Kar5 protein was included as an outgroup. Numbers at nodes indicate bootstrap values calculated from 1000 replicates. The tree is drawn to scale, with branch lengths reflecting the number of substitutions per site. Scale bar, 0.3 substitutions per site.

**Table 1 plants-12-03608-t001:** Nuclear fusion factors conserved between *Saccharomyces cerevisiae* and *Arabidopsis thaliana*.

	*Saccharomyces cerevisiae*	*Arabidopsis thaliana*
BiP	BiP/Kar2	BiP
ER-resident J proteins	Sec63, Jem1/Kar8	ERdj3A, ERdj3B, P58^IPK^
GEX1/Kar5 family	Kar5	GEX1
SUN protein	Mps3	SUN proteins

## Data Availability

Not applicable.

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
