# Peer review of "Nuclear Fusion in Yeast and Plant Reproduction"

_plants, 2023, doi:10.3390/plants12203608_

Round 1

Reviewer 1 Report

Kobayashi and Nishikawa summarized recent research on the nuclear fusion process during plant sexual reproduction in comparison with the nuclear fusion process during the mating of the budding yeast. The authors first summarized knowledge from yeast genetics to introduce genes involved in the nuclear congression process and the nuclear membrane fusion process. Then, the author summarized the knowledge on the nuclear fusion process during plant sexual reproduction with particular focus on function of ER-localized molecular chaperons and other nuclear membrane proteins such as SUN-KASH and GEX1. In the end, the authors discuss roles of nuclear fusion in development of embryos and endosperms. Overall, the manuscript is well organized, written in clear English and comprehensively and concisely summarizes recent progress in the field. I have only minor comments below which would help the author to improve the manuscript.

1. The title of the paper reflects only sections describing about plant reproduction. It would be better to change the title to reflect the yeast part and conservation of molecular mechanisms of nuclear fusion between plants and the yeast.

2. Line 79: What is Mps2? Does it belong to a specific family of proteins?

3. Line 79-80: “Mutants of these proteins are defective in nuclear fusion” I wonder whether they show specific defects in the nuclear congression process, since the section is focusing on the nuclear congression process.

4. Line114-115: What kind of protein does the PRM3 gene encode?

5. Line 189-192: the first sentence says that the erdj3a p58ipk mutant is defective in sperm nuclear fusion, but the following sentence says that the erdj3a p58ipk mutant is not defective in sperm nuclear fusion. Which one is correct? I think that one of them would be the erdj3b p58ipk mutant?

6. Line 205-207: “Expression of SUNDN in developing female gametophytes resulted in polar nuclear fusion, indicating the roles of SUN proteins in this process.”

Would you mean polar nuclear fusion “defect”?

7. Line 214-221: The reference No. 23 should be referred here.

8. Line 238-240: It would be better to write full species names of Zebrafish, Chlamydomonas and Plasmodium.

Author Response

Thank you very much for your comments on our manuscript. Your comments are favorable and constructive. Text changes are highlighted in yellow. Here are our point-by-point responses to your comments.

1. The title of the paper reflects only sections describing about plant reproduction. It would be better to change the title to reflect the yeast part and conservation of molecular mechanisms of nuclear fusion between plants and the yeast.

According to the reviewer’s suggestion, we changed the title of this manuscript to “Nuclear Fusion in Yeast and Plant Reproduction.”

2. Line 79: What is Mps2? Does it belong to a specific family of proteins?

Mps2 is an integral membrane protein localized in the outer nuclear membrane. We added this description to the revised manuscript.

3. Line 79-80: “Mutants of these proteins are defective in nuclear fusion” I wonder whether they show specific defects in the nuclear congression process since the section is focusing on the nuclear congression process.

According to the reviewer’s comment, we changed the sentence to “Mutants of these proteins are defective in the nuclear congression process.”

4. Line114-115: What kind of protein does the PRM3 gene encode?

Prm3 is a pheromone-induced peripheral membrane protein on the cytoplasmic face of the outer nuclear membrane. We added the description to the revised manuscript.

5. Line 189-192: the first sentence says that the erdj3a p58ipk mutant is defective in sperm nuclear fusion, but the following sentence says that the erdj3a p58ipk mutant is not defective in sperm nuclear fusion. Which one is correct? I think that one of them would be the erdj3b p58ipk mutant?

The latter should be erdj3b p58ipk. We corrected the text. Thank you very much for your comment.

6. Line 205-207: “Expression of SUNDN in developing female gametophytes resulted in polar nuclear fusion, indicating the roles of SUN proteins in this process.” Would you mean polar nuclear fusion “defect”?

Yes, “polar nuclear fusion defect” is correct. We corrected the sentence. Thank you very much.

7. Line 214-221: The reference No. 23 should be referred here.

We added the reference accordingly. Since we added two papers to the reference list of the revised manuscript, No. 23 changed to No. 24.

8. Line 238-240: It would be better to write full species names of Zebrafish, Chlamydomonas and Plasmodium.

According to the reviewer’s suggestion, we added full species names to these organisms.

Reviewer 2 Report

In this short article, the authors summarized the genes and proteins involved in nuclear fusion in yeast and A. thaliana. The identified proteins revealed surprising conservations of nuclear fusion mechanisms between yeast and plants. However, all these proteins are identified via genetic analysis, lacking molecular mechanisms and the manuscript is quite short, which may be more suitable to publish as a short communication rather than a review. Nevertheless, the content is interesting and the authors' summary is clear. I did not find any serious problems with this manuscript. The only issue I find is on Line 58 "MATa and MATa", one of the "MATa" should be an alpha symbol rather than the letter "a" according to Figure 2A.

Author Response

Thank you very much for your comments on our manuscript. Your comments are favorable and constructive. Text changes are highlighted in yellow. Here are our point-by-point responses to your comments.

In this short article, the authors summarized the genes and proteins involved in nuclear fusion in yeast and A. thaliana. The identified proteins revealed surprising conservations of nuclear fusion mechanisms between yeast and plants. However, all these proteins are identified via genetic analysis, lacking molecular mechanisms and the manuscript is quite short, which may be more suitable to publish as a short communication rather than a review. Nevertheless, the content is interesting and the authors' summary is clear. I did not find any serious problems with this manuscript.

According to the reviewer’s suggestion, we changed the article type of this manuscript to Communication.

The only issue I find is on Line 58 "MATa and MATa", one of the "MATa" should be an alpha symbol rather than the letter "a" according to Figure 2A.

Yes, alpha is correct. We corrected the text accordingly.